# Characteristics of Microstructural Changes Associated with Glioma Related Epilepsy: A Diffusion Tensor Imaging (DTI) Study

**DOI:** 10.3390/brainsci12091169

**Published:** 2022-08-31

**Authors:** Hong Zhang, Chunyao Zhou, Qiang Zhu, Tianshi Li, Yinyan Wang, Lei Wang

**Affiliations:** 1Department of Neurosurgery, Beijing Tiantan Hospital, Capital Medical University, Beijing 100070, China; 2Beijing Neurosurgical Institute, Capital Medical University, Beijing 100070, China

**Keywords:** glioma-related epilepsy, diffusion tensor imaging, tract-based spatial statistical

## Abstract

(1) Background: Glioma is the most common primary tumor in the central nervous system, and glioma-related epilepsy (GRE) is one of its common symptoms. The abnormalities of white matter fiber tracts are involved in attributing changes in patients with epilepsy (Rudà, R, 2012).This study aimed to assess frontal lobe gliomas’ effects on the cerebral white matter fiber tracts. (2) Methods: Thirty patients with frontal lobe glioma were enrolled and divided into two groups (Ep and nEep). Among them, five patients were excluded due to apparent insular or temporal involvement. A set of 14 age and gender-matched healthy controls were also included. All the enrolled subjects underwent preoperative conventional magnetic resonance images (MRI) and diffusion tensor imaging (DTI). Furthermore, we used tract-based spatial statistics to analyze the characteristics of the white matter fiber tracts. (3) Results: The two patient groups showed similar patterns of mean diffusivity (MD) elevations in most regions; however, in the ipsilateral inferior fronto-occipital fasciculus (IFOF), superior longitudinal fasciculus (SLF), and superior corona radiata, the significant voxels of the EP group were more apparent than in the nEP group. No significant fractional anisotropy (FA) elevations or MD degenerations were found in the current study. (4) Conclusions: Gliomas grow and invade along white matter fiber tracts. This study assessed the effects of GRE on the white matter fiber bundle skeleton by TBSS, and we found that the changes in the white matter skeleton of the frontal lobe tumor-related epilepsy were mainly concentrated in the IFOF, SLF, and superior corona radiata. This reveals that GRE significantly affects the white matter fiber microstructure of the tumor.

## 1. Introduction

Glioma is the most common primary tumor in the central nervous system, and glioma-related epilepsy (GRE) is one of the resulting common symptoms [1]. Preoperative seizures are an important indicator of glioma diagnosis and progression [1]. Additionally, perioperative epilepsy control is also significantly associated with the prognosis of glioma patients. Among the indicators, lower WHO levels of glioma [2], the specific tumor location [3], subtypes of mutant IDH typing glioma [4,5], and other clinically relevant indicators are closely related to preoperative epilepsy [6,7]. The occurrence of glioma related epilepsy is associated with white matter network integrity [6], and may have concurrent structural and connectivity abrasions of brain areas distant from the brain tumor [7]; however, there have been few systematic studies on the location of tumor epilepsy susceptibility areas. Only a few retrospective studies have targeted the susceptible brain regions of tumor-related epilepsy, and the results differed. Gonen et al. found that tumors involving the supplementary motor area (SMA) are prone to epilepsy during awake craniotomy [3,8]. Another multi-center big data study also found that parietal and insular lobe involvement were independent predictors of tumor epilepsy [9]. Unfortunately, the above studies did not accurately locate and quantify epileptogenic tumors. In the only two voxel-based lesion-symptom mappings (VLSM) studies for glioma-associated epilepsy-susceptible areas, Wang et al. confirmed that the tumors involved in the left anterior motor region were more susceptible to causing GRE [3], and Yunhe et al. also found that the bilateral frontal lobe is an area susceptible to GRE [5,10]. Although the above studies reported different results in tumor-associated epilepsy susceptibility areas, it is consistent that the GRE areas are primarily in the frontal lobe. The extensive data survey also found that the incidence of preoperative epilepsy in diffuse low-grade glioma of the frontal lobe is also very high (87.6%) [9]. Furthermore, the correlation mechanisms between epilepsy and susceptible brain regions have not been carefully scrutinized. We suppose that it has to be something different in the cortical or subcortical structures of the brain which causes the discrepancy in seizure occurrence. 

It is widely believed that GRE is caused by the invasion and destruction of subcortical structures, leading to a whole brain functional disorder [9,11]. Previous research has confirmed multiple global network attribute changes in patients with epilepsy [12,13,14]. The structural networks are also involved and are related to abnormalities in the main white matter fiber tracts in patients with epilepsy [15,16]. These abnormalities are not only located near the tumor or epileptic foci but also on the distal side [17,18]; however, previous studies on white matter structure have primarily focused on patients with non-neoplastic epilepsy. Consequently, there are few studies on the changes in white matter fiber tracts in tumor-associated epilepsy; therefore, key questions remain unanswered. Do frontal glioma-related epilepsy patients have distant white matter structure changes? Are those changes related to the occurrence and development of epilepsy? All these questions remain unanswered.

Diffusion tensor imaging (DTI) is an important method to evaluate the structure of white matter structures. Diffusion is sensitive to water movement, particularly along axonal fibers. DTI provides useful information about diffusion measurements and enables the calculation of several parameters from DTI [19]. The four diffusion coefficients, including fractional anisotropy (FA), mean diffusivity (MD), axial diffusivity (AD), and radial diffusivity (RD), are derived from mathematical modeling and provide information and indicators when assessing the imaging studies of glioma. The study of glioma-DTI found significant differences in FA, MD, and other DTI coefficients in many white matter regions where no signal changes in conventional MRI were found [17,18]. Previous studies have suggested that these changes are related to tumor invasion and development [18], and more emphasis is placed on the analysis of the dispersion values of FA, MD, and others of one or more white matter fiber pathways, but lacking an overall, voxel-wise data inspection. The tract-based spatial statistics (TBSS) analysis technique [20] can be used to test the difference in the major white matter pathways of the whole brain at the voxel level. The parameters of fibers, such as FA and MD, improve the accuracy and reliability based on a traditional fiber bundle analysis.

This study uses TBSS to identify the microstructural abnormalities of white matter fiber tracts of frontal lobe glioma with GRE by combining a conventional magnetic resonance and DTI sequence and to explore the whole brain level white matter fiber changes in patients with frontal GRE; therefore, with these investigations, we aim to provide evidence for further study of the mechanism of GRE.

## 2. Materials and Methods

The study was conducted according to the guidelines of the Declaration of Helsinki, and approved by the Institutional Review Board of Beijing Tiantan Hospital. Written in- formed consents have been obtained from the patients involved in the study to publish this paper.

### 2.1. Participants

A group of 30 patients pathologically diagnosed with frontal glioma in Beijing Tiantan Hospital between 2014 and 2017 was originally enrolled in this study. Among them, 5 patients were excluded due to apparent insular or temporal involvement. Thus, 25 patients (12 males and 13 females, median age (years) = 42.04 ± 9.5) with either a right (*n* = 12) or left (*n* = 13) frontal glioma were finally enrolled in Appendix A. Among them, 12 patients had had postsurgical seizure onset (L:R = 6:6) while the other 13 (L:R = 6:7) had not, thereby composing the EP group and nEP group, respectively. A set of 14 age and gender-matched healthy controls were also included. All the enrolled subjects underwent conventional magnetic resonance imaging (MRI) and DTI preoperatively. Basic clinical data and neurofunctional status (including gender, age, language, and cognition) of the patients were collected. All patients underwent a tumor resection and were pathologically diagnosed as WHO grade II-IV glioma. This study was approved by the Institutional Review Broad of Beijing Tiantan Hospital, and written informed consent was provided by all the study participants.

Glioma-related epilepsy is mainly diagnosed based on symptoms. For people with an abnormal EEG but no seizures, we cannot diagnose GRE. Furthermore, the false-negative rate of an EEG itself is high, and even if there is no abnormality on the EEG, an epileptic discharge may still be detected in the cortical EEG during surgery; therefore, in order to unify the inclusion criteria, we did not include occult epilepsy in the Ep group.

All epilepsy symptoms were evaluated in accordance with the IALE guidelines. All patients in the epilepsy group had at least one seizure occurrence before the imaging examination.

### 2.2. MRI Acquisition

MRI was performed with a MAGNETOM Prisma 3T MR scanner (Siemens, Erlangen, Germany). The T1 and T2 weighted image parameters were as follows: TR = 5800 ms, TE = 110 ms, flip angle = 150°, FOV = 230 × 230 mm, and voxel size = 0.6 × 0.6 × 5.0 mm^3^. The diffusion tensor images were acquired using a single shot, echo planar imaging sequence (axial slices = 75, resolution = 2.0 × 2.0 × 2.0 mm, TR = 6000 ms, TE = 103 ms, FOV = 230 × 230 mm, different directions = 30, b = 0/1000 s/mm^2^, and EPI factor = 154).

### 2.3. Tumor ROI Extraction

The region of tumors of each patient was semi-automatically created using the T2 weighted image using MRIcron software by two experienced neurosurgeons. T2 weighted images with abnormal hypertense signals were all determined as tumor areas, while the cerebrospinal fluid was carefully avoided as this part may cause unmasking of the tumor margins. The lesion map would be re-evaluated by an experienced neuro-radiologist afterward, and if the discrepancy was >5%, the final determination would be up to the radiologist. All tumor masks were registered to the MNI152 standard brain template with FSL, and the left-sided tumor masks were all Y-axis flipped. Every standardized tumor mask was transferred to FSL Maths to evaluate the tumor volume. Finally, all the tumor masks were overlapped to generate a lesion map in the standard space. A binary mask of this lesion map was extracted with FSL Maths.

### 2.4. Diffusion Data Processing

Preprocessing and analysis of the diffusion dataset were implemented using a pipeline toolbox, PANDA (http://www.nitrc.org/projects/panda/, accessed on 3 March 2012), which was developed based on the FMRIB software library (FSL, https://fsl.fmrib.ox.ac.uk/fsl/fslwiki/, accessed on 3 March 2012), the Pipeline system for Octave and MATLAB (PSOM), the Diffusion Toolkit (http://www.trackvis.org/, accessed on 3 March 2012) and MRIcron (http://www.mccauslandcenter.sc.edu/mricro/mricron, accessed on 3 March 2012). The data processing steps and TBSS analysis have been clarified explicitly by Cui [21]. First, steps to extract the basic DTI metrics were implemented, including extracting the brain mask, correcting the eddy current effect, averaging multiple acquisitions, calculating the diffusion tensor, and producing metrics. Before the TBSS process was implemented, the standardized and resliced FA and MD maps of each individual with left-sided tumors were all flipped among the Y axis, resulting in all patients having occupations on the same side. After this step, the TBSS process consisted of aligning each subject’s FA image, creating the mean FA map, extracting the FA skeleton, and projecting the individual subjects’ FA onto the skeleton [22].

Before the statistical analysis, we implemented a Y-axis flip procedure among the standardized tumor masks and FA, with MD maps in 12 patients with left-sided gliomas and their matched controls, causing all the tumors to be located in the right hemisphere. Following these procedures, an overlapped tumor mask was generated (Figure 1), and every individual’s data within this mask was wiped out. Thus, our results could be simply defined as ipsi-tumoral fiber changes or contra-tumoral fiber changes.

### 2.5. Statistical Analysis

The clinical characteristics were compared between the EP group, nEP group, and controls using a one-way ANOVA and Fisher’s exact test. As for comparisons of the atlas-based results, an unpaired *t*-test was performed to calculate the differences between EP vs. the controls and nEP vs. the controls. A voxel-wise analysis was carried out utilizing the general linear model (GLM) with the FSL randomize tool [22], using a 5000 times permutation test within the lesion-excluded average FA skeleton mask between the EP/nEP groups and healthy controls. Significant clusters were defined with the threshold-free cluster enhancement method [15], and the resulting *p*-value maps were corrected by family-wise error (FWE) at the TFCE level with *p* < 0.05. The cluster locater tool was used to locate the significant clusters in a specific fiber-tract in the ICBM-DTI-152 white matter atlas. Single masks of the alternated fiber clusters in a group (e.g., LIG patients, and RIG patients vs. controls) were created and used to extract the mean FA and MD values for every subject using PANDA mask extractor tools.

## 3. Results

### 3.1. Demographic Characteristics

The clinical characteristics of all enrolled patients are summarized in Table 1. Briefly, there were 12 patients in the EP group and 13 in the nEP group with either left- or right--sided frontal glioma. The number of lower-grade gliomas (WHO II-WHO III) and GBM (WHO IV) were 20 and 5, respectively. A total of seven patients (LIG = 4, and RIG = 3) had a motor defect including limbal weakness and facial paralysis. Four patients (LIG = 3, and RIG = 1) had language defects, and five (LIG = 3, and RIG = 2) patients experienced tumor-related cognitive defects, such as memory deterioration, slowness in reacting and disorientation. The average tumor volumes of each cohort were 50.88 ± 40.48 mm^3^ and 27.76 ± 17.36 mm^3^ (EP and nEP, respectively). The *t*-test between the two groups showed that the differences were not significant for the tumor volume (*p* = 0.25). There were no significant differences with regard to age, sex, WHO grade, pre-operational epilepsy, motor defect, sensor defect, language defect, and cognitive defects between the EP and nEP groups (Fisher’s exact test, *p* < 0.05).

### 3.2. Atlas-Based General Analysis of Diffusion Parametric

Significant alternations (FWE corrected, *p* < 0.05) of the fractional anisotropy (FA) and mean diffusivity (MD) values were observed in both of our group comparisons (EP vs. control and nEP vs. control). Both patient groups had degenerated FA and elevated MD. Since the tumor area and the peritumoral edema zone were all ruled out from the analysis, in both tests, we only found several tiny (5–15 voxels) clusters with FA degeneration (Table 1). Those clusters were considered negligible due to their small size and far separation of each cluster; however, widespread changes in the MD with either ipsi-tumoral and contra-tumoral large fiber tracts were detected in both tests (Table 2 and Table 3). The two patient groups showed similar patterns of MD elevations in most regions; however, in the ipsilateral IFOF, SLF, and superior corona radiata, the significant voxels of the EP group were more than the nEP group (Figure 1). No significant FA elevations or MD degenerations were found in the current study.

### 3.3. TBSS-EP Group versus Control

In the voxel-wise permutation test between the EP patients and controls, significant mean diffusivity elevations were detected in several ipsi-tumoral and contra-tumoral major fiber bundles. Only fibers with a cluster size of more than 50 voxels were displayed in our results (Table 2). The alternated ipsi-tumoral tracts were: superior longitudinal fasciculus (voxel size = 1755), superior corona radiata (voxel size = 1342), inferior fronto-occipital fasciculus (voxel size = 488), and body of corpus callosum (voxel size = 900). The contra-tumoral MD elevated tracts were superior longitudinal fasciculus (voxel size = 625), anterior corona radiata (voxel size = 1577), superior corona radiata (voxel size = 1150), and posterior thalamic radiation (voxel size = 481).

### 3.4. TBSS-nEP Group versus Control

The EP group showed similar patterns in many brain regions to the nEP group at the MD alternations (Figure 1); however, the differences between the two comparisons were obvious too. The voxel size of the SLF (VS = 1755), IFOF (VS = 488), and superior.corona radiata (VS = 1342) of the ipsi-tumoral side was larger than the nEP group (Figure 2). Except for those tracts, other tracts of the EP group with alternated MD values were close to the nEP group at either the voxel size or spacial distribution, including ipsi-tumoral Cingulum (VS = 279 and 243, respectively), and ipsi-tumoral uncinate.fasciculus (VS = 46 and 32).

## 4. Discussion

In this study, we used TBSS to generate a white matter skeleton in frontal lobe glioma patients with tumor-related epilepsy. We found that patients with GRE can experience micro-structural changes in the white matter fiber skeleton in the extra-tumoral area. The ipsilateral superior corona radiata, superior longitudinal fasciculus, and inferior fronto-occipital fasciculus had significantly higher MD values than the controls, and these changes did not appear in the non-epileptic groups. These results indicate that the epilepsy group had more disruption in the fiber tracts, which can be used as an effective means to monitor the occurrence and progression of glioma-related epilepsy.

We used FA and MD data to compare the normal controls and epilepsy or non-epilepsy patient groups. Among them, the comparison of the MD values showed widespread significant clusters (Table 3). In contrast, both comparisons only resulted in a few significant clusters when suing FA for comparison. Because these clusters were too small (vs. <20) and the distribution was too scattered, we believe they cannot represent the overall change of the fiber tracts.

Previous studies on DTI in patients with epilepsy have shown that MD may be a more accurate quantitative indicator, which is consistent with the current study [23]. In particular, the MD decreased in TLE patients while the distance between the fiber tracts and the temporal lobe became longer. In an analysis of medial temporal lobe epilepsy [24], MD significantly increased ipsilaterally and contralaterally in most white matter regions; therefore, the white matter network of the ipsilateral brain was more severely interfered with [25], indicating that these changes have a localizing effect.

Meanwhile, studies have found that patients with left-sided TLE had a negative correlation between the time from last seizure and MD in the ipsilateral distal portion of the white tracts [26], which indicates that changes in the white matter integrity may gradually develop during epilepsy. Another previous DTI study of temporal lobe epilepsy suggested that the area of changed MD value was closer to the location of epilepsy-like discharge on an EEG [24,25,26,27,28,29,30,31]; therefore, an MD increase in multiple white matter regions can accurately describe the epilepsy-related tract abnormalities [25], which is consistent with our research results. In conclusion, MD changes are more specific in predicting potential epileptic foci [25].

Due to the space-occupying effect of a tumor, the corresponding fiber tracts at the tumor border and surrounding tissues are pushed, compressed, infiltrated, or destroyed. Previous studies have shown that the tumor mass will affect the FA or MD values of the tumor area and the surrounding area of the tumor [23]. GBM infiltration and edema are usually related to decreased FA, and a study comparing radiation damage and tumor infiltration found that the apparent diffusion coefficient (EEG) in regions of a tumor infiltration were reduced when compared to radiation damage, which was related to an excessive number of tumor cells [28]. Notably, a study comparing radiation damage and tumor infiltration found that ADC in the regions of tumor infiltration was reduced compared to radiation damage, which was related to an excessive number of tumor cells [28].This has led some researchers to use DTI image data to help detect tumor boundaries and infiltration [29,30]. The mass-occupying effect may directly lead to an abnormal white matter skeleton. In our study, to verify the impact of tumor-related epilepsy on fiber bundles, the effect of a tumor direct mass effect needs to be excluded. To this end, we performed some necessary preprocessing before calculating the white matter skeleton TBSS. On the one hand, we marked the tumor area of each patient, superimposed between groups, and took the union of all tumors in the group to remove. This operation ensured that the tumor did not directly affect all the calculated fiber tracts. On the other hand, lesions could have caused micro-structural changes in non-tumor areas on the ipsilateral side and even on the contralateral side [31,32,33]. Notably, if the GRE group was compared with the non-epilepsy group TBSS, this may have resulted in false-negative results due to the effect of tumor occupation [34]; therefore, we separately compared the white matter fiber skeleton of the epilepsy group and the non-epilepsy group with the normal group, and then found the difference between the two groups of results. Thus, we could measure and distinguish the effect caused by glioma and GRE.

Changes in the MD value of the white matter skeleton are caused by tumor-related secondary epilepsy. It is known that an increased MD is due to a reduction in the number and/or volume of cells and a compensatory increase in the extracellular space of WM [35].

In our study, the extra-lesion region on the ipsilateral and the contralateral hemisphere showed a certain degree of consistency. We considered that this similar result might have been caused by the mass effect of the tumor itself on the white matter abnormalities (Table 4), which include ipsi-tumoral cingulum and ipsi-tumoral uncinate fasciculus.

At present, some studies on frontal lobe focal motor epilepsy have shown that the FA and MD values of the bilateral suboccipital frontal tract, bilateral anterior thalamus radiation, bilateral corticospinal tract, and bilateral inferior fronto-occipital fasciculus have abnormal changes [36]. In our study, the abnormalities in the ipsi-tumoral cingulum and uncinate fasciculus fiber tracts were consistent with the former report, except for the frontal part of the corpus callosum. Since the increase in the MD value of the fiber tracts indicates the destruction of the sphingomyelin layer or a decrease in the axon density [36,37], the MD indicates the viscosity of the fluid and is sensitive to white matter edema or damage [38]. It is known that such damage to the white matter skeleton plays an important role in the occurrence of epilepsy [39,40,41,42]; therefore, we consider that the occurrence of frontal lobe epilepsy related to glioma might be associated with damage to the fiber tracts above.

## 5. Limitation

The current research used TBSS to explore microstructural white matter alterations associated with GRE. The results clarified a potential invasive pattern of tumors; however, some limitations must be considered. Due to technological reasons, we only used DTI for the analysis; therefore, structural and functional inter-group differences in the cortex may have been missed. We were unable to carry out a voxel-wise analysis considering the functional deficits and specific white matter structure for the current dataset. Only frontal lobe glioma was included in this study to maintain the consistency of the tumor location; therefore, the network change of glioma originating from other brain locations remains unclear. The number of enrolled patients was too small to compare subgroups, such as patients with different tumor grades or with or without cognitive damage. Our future research will focus on expanding the dataset. Thus, tumors of other types or the same type in different locations could also be studied with the enrollment of a sufficient number of patients. Furthermore, multi-model analyses with connectome construction and graph-theory analyses could clarify the effects of GRE.

## 6. Conclusions

This study used DTI in the TBSS of the epilepsy patient group and the non-epilepsy patient group compared with the normal control group. It was found that the changes in the white matter skeleton of the frontal lobe tumor-related epilepsy were mainly concentrated in the ipsilateral inferior fronto-occipital fasciculus (IFOF), superior longitudinal fasciculus (SLF), and superior corona radiata. This reveals the abnormality of the white matter structure in TBSS, which is outside the tumor area, and the change of TBSS can be used as a predictor of tumor-related epilepsy, predicting the prognosis of patients with epilepsy after a simple tumor resection before surgery; therefore, this may have a specific guiding role in the prognosis of postoperative epilepsy.

## Figures and Tables

**Figure 1 brainsci-12-01169-f001:**
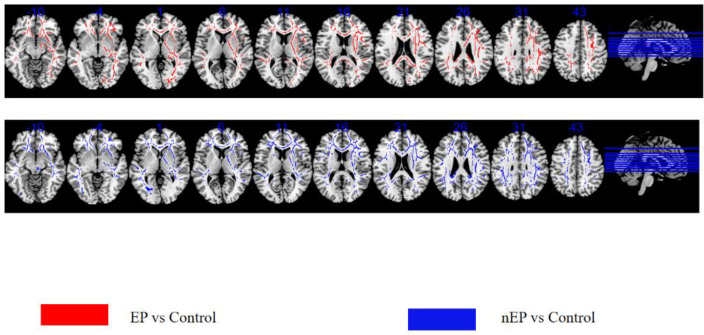
Significant clusters were identified by permutation testing (*n* = 5000, and FWE-corrected at TFCE level *p* < 0.05).

**Figure 2 brainsci-12-01169-f002:**
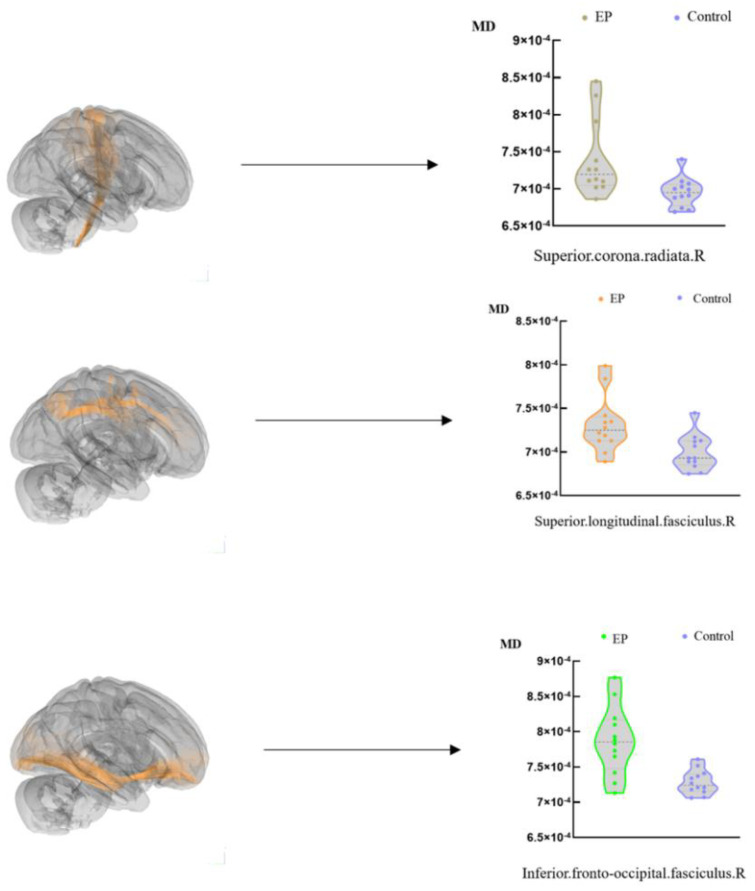
Atlas−based analysis showed significant MD increases in the EP group of glioma patients (*T*-test, and FWE-corrected *p* < 0.05).

**Table 1 brainsci-12-01169-t001:** Clinical characteristics of patients.

	Ep	nEp	Con	*p*-Value
Age range (mean ± SE)	43.92 ± 10.82	45.00 ± 9.03	44.85 ± 3.31	0.94 *
Sex (female/male)	8/4	6/7	6/7	0.77 ^^^
Histology (HGG/LGG)	3/9	5/8	/	0.91 ^^^
Tumor volume (mL)	50.88 ± 40.48	27.76 ± 17.36	/	0.25 ^u^

Abbreviations: HGG = high-grade glioma; LGG = low-grade glioma. *: one-way ANOVA; ^^^: Fisher’s exact test; ^u^: unpaired *t*-test.

**Table 2 brainsci-12-01169-t002:** Significant clusters with the most voxel count of mean diffusivity.

Cluster Location	Voxel Size of Mean Diffusivity	
	Ep	nEp
Superior.longitudinal.fasciculus.R	1755	1577
Superior.corona.radiata.R	1342	1150
Inferior.fronto-occipital.fasciculus.R	488	481

Note: The numbers represent the voxel count of the cluster, cluster-wise FEW; R means the ipsilateral side of the tumor; L means the contralateral side of the tumor.

**Table 3 brainsci-12-01169-t003:** Atlas-based results.

Cluster Location	Atlas-Based Results		*p*-Value	
	Ep	nEP	Ep vs. Control	nEp vs. Control
Superior.longitudinal.fasciculus.R	0.4626 ± 0.0286	0.4565 ± 0.0304	0.03 ^u^	0.026 ^u^
Superior.corona.radiata.R	0.4813 ± 0.0270	0.2484 ± 0.0310	0.028 ^u^	0.012 ^u^
Inferior.fronto-occipital.fasciculus.R	0.4479 ± 0.0590	0.4861 ± 0.0193	0.034 ^u^	0.025 ^u^

^u^: unpaired *t*-test.

**Table 4 brainsci-12-01169-t004:** Significant clusters of mean diffusivity.

Cluster Location	Voxel Size of Mean Diffusivity	
	Ep	nEp
Cingulum.(cingulate.gyrus).R	279	243
Uncinate.fasciculus.R	46	32

## Data Availability

Anonymized data and material will be available on reasonable request.

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
