# Peer review of "Characteristics of Microstructural Changes Associated with Glioma Related Epilepsy: A Diffusion Tensor Imaging (DTI) Study"

_brainsci, 2022, doi:10.3390/brainsci12091169_

Round 1

Reviewer 1 Report

"Characteristics of microstructural changes associated with related epilepsy of glioma patients: a DTI study" is an interesting article. The aim was to identify microstructural abnormalities of white matter fiber tracts of frontal lobe glioma with glioma related epilepsy by combining the conventional magnetic resonance sequence and DTI sequence, and to explore the changes the whole brain level white matter fiber changes in patients with frontal glioma-related epilepsy. The results shown that patients with glioma related epilepsy can cause micro-structural changes in the white matter fiber skeleton in the extra-tumoral area. The ipsilateral superior corona radiate, superior longitudinal fasciculus and inferior fronto-occipital fasciculus had significantly higher MD values compared to the controls; these changes didn’t appear in non-epileptic groups.

There are many issues in the article that need to be addressed.

Abstract 

The abstract has confusing wording, for example the last paragraph is an exact copy of the conclusion section and therefore does not fit the structure that an Abstract should have. For example, the term TBSS is not defined. The abbreviations IFOF and SLF are used on line 19 but are defined up to line 25. Careful and complete rewriting of this section is suggested. 

Introduction 

Line 46. Define VLSM.

Lines 69 and 70 define the parameters used in DTI, but line 254 of the Discussion uses ADC which is not defined in this part. In addition, it is suggested to briefly explain the biological meaning of each of these parameters.  

Materials and Methods 

In line 92 it is suggested to add the standard deviation in addition to the mean age. 

Line 94 EP group and nEP group. There should be consistency in the abbreviations of these groups throughout the article. For example, in Table 2 Ep and Nep are used. The criteria used to form these groups are not mentioned, how was the presence of seizures determined clinically by EEG? A control group is also mentioned but the characteristics of said group are not described. 

Line 143. 2.5. Statistical analysis. It is suggested to add the statistical analysis used for the different demographic variables and for the conventional MRI and DTI, in the results section the Fisher test and the Student's t test are mentioned. This section is appropriate for the description of such statistical tests.  

Lines 153 to 155. The paragraph “Interventional studies involving animals or humans, and other studies that require ethical approval, must list the authority that provided approval and the corresponding ethical approval code” is out of context, what is it referring to?  

Results 

It is suggested to review the entire section and rewrite it, in its current form it is confusing. On line 165 add the standard deviation. In line 166 “no significant result” it is mentioned that the differences were not significant.  

Table 1 is very scattered, has inconsistent spaces, is poorly aligned, and there is no total column with the mean values ​​and standard deviation.

In Table 2. The names of the groups are in lower case. Mean values, their standard deviation, and statistical comparison between the two groups and p-values ​​are not shown.     

Discussion  

In line 230 of it refers to a Table 3 that does not exist

Line 237. Why “META analysis” with capital letters? 

Line 254. The abbreviation ADC is not defined.  

References 

A careful review of all references is suggested. There is inconsistency in the way journal names are spelled, for example Brain or Brain: a journal of neurology. In some citations the volume number is given and in others it is not. Etc. 

Author Response

Dear reviewer:

    We would like to thank you for your careful reading, helpful comments, and constructive suggestions, which has significantly improved the presentation of our manuscript.

     We have carefully considered all comments from the reviewers and revised our manuscript accordingly. The manuscript has also been double-checked, and the typos and grammar errors we found have been corrected. In the following section, we summarize our responses to each comment from the reviewers. We believe that our responses have well addressed all concerns from the reviewers. We hope our revised manuscript can be accepted for publication.

Reviewer 2 Report

1. It is advised to avoid abbreviations in the title. Moreover, the title should mention the location of the study and the type of the study. E.g. “DTI”

2. Methods

2.1 IRB number is mandatory.

2.2 A table with baseline characteristics of the individual of the study (including control) should be provided.

2.3 Who did the author diagnose these individuals' pathologies like epilepsy and glioma?

2.4 What were the diagnosis criteria for glioma? The authors should use the last classification system.

2.5 All the abbreviations including imaging techniques should be fully described at the first presentation.

3. Statistics. Statistics are poorly described.

3.1 Describe variable distributions

3.2 Is this a preliminary study? Describe the power of the study.

3.3 Describe the tests used accordingly: parametric, nonparametric

3.4 If we have three groups, why did the authors not perform ANOVA?

3.5 How did the authors exclude confounding variables?

3.6 Provide a type of correlation.

4. Result

4.1 Table 1. Why patients have this nomenclature Sub010, Sub018

5. Discussion

5.1 The author should provide limitations of the study

In line 153, the following structure is misplaced “Interventionary studies involving animals or humans, and other studies that require ethical approval, must list the authority that provided approval and the corresponding ethical approval code”

Author Response

 Dear Reviewers:

    We would like to thank you for your careful reading, helpful comments, and constructive suggestions, which has significantly improved the presentation of our manuscript.

    We have carefully considered all comments from the reviewers and revised our manuscript accordingly. The manuscript has also been double-checked, and the typos and grammar errors we found have been corrected. In the following section, we summarize our responses to each comment from the reviewers. We believe that our responses have well addressed all concerns from the reviewers. We hope our revised manuscript can be accepted for publication.

Reviewer 3 Report

Thanks for recommending me as a reviewer. In this paper, authors aimed to assessing the effects of frontal lobe gliomas on the cerebral white matter fiber tracts. In this paper, thirty patients with frontal lobe glioma were enrolled and divided into two groups. A set of age and gender matched healthy controls was also included. All the enrolled subjects underwent conventional MR images and DTI preoperatively. In this paper, authors used tract-based spatial statistics to analyze the characteristics of white matter fiber tracts. If authors complete minor revisions, the quality of the study will be further improved. 

1. The introduction section is well written. If the authors describe in more detail the trends in previous studies related to microstructural changes associated with related epilepsy of glioma patients in the introduction section, it can help readers understand.

2. line 199: The title and content of the table must be presented on the same page.

3. It would be helpful to the reader if the authors add some limitations to the discussion section.

4. It would be helpful to the reader if the authors add implications for future research in the Conclusion section.

Author Response

(The authors gave the same response as above.)

Reviewer 4 Report

Comments attached

Author Response

(The authors gave the same response as above.)

Round 2

Reviewer 1 Report

The new version of the paper "Characteristics of microstructural changes associated with related epilepsy of glioma patients: a DTI study" has many of the suggested changes and adequate argumentation for them. The changes made have improved the quality of paper.

Author Response

Dear reviewer:

Thank you for your encouraging comment!

Reviewer 2 Report

1. The title lacks the type of the study and the location where the study was done.

Altman DG, Simera I, Hoey J, Moher D, Schulz K. EQUATOR: reporting guidelines for health research. Open Med 2008;2:e49-50.

2. Revise authors' filiation numbers.

3. It is advised to provide all pieces of evidence as a possibility. It is worth remembering that health sciences are not exact.

E.g

"The abnormalities of white matter fiber tracts are involved in attributing"

The abnormalities of white matter fiber tracts are probably involved in …

4. The authors should upload supplementary material or include the methods.

How were the patients diagnosed with epilepsy? Have the patients performed video-EEG?

Did all the patients do a brain biopsy for the final diagnosis of glioma?

Was further assessed the glioma-type?

What was the protocol for the selection of the patients?

5. The statistical and methods queries discussed in the first round should be included in the manuscript. E.g., Variable distribution, IRB number, tests used, study power, correlation type, and confounding variables assessment.

6. Table 3. It is advised to revise the symbol "$."

7. Could the authors explain why the controls are healthy individuals? If they assess epilepsy in a secondary pathology, the controls should have the same pathology to minimize confounding factors. Otherwise, the results could be significant, and managing the high number of variables will be challenging.

8. It would be interesting to include a table in the discussion about the studies with voxel-based lesion-symptom mappings provided in the introduction.

Wang Y, Qian T, You G, Peng X, Chen C, You Y, Yao K, Wu C, Ma J, Sha Z, Wang S, Jiang T. Localizing seizure-susceptible brain regions associated with low-grade gliomas using voxel-based lesion-symptom mapping. Neuro Oncol 2015;17:282-8.

Yunhe M, Yuan Y, Xiang W, Yanhui L, Qing M. Mapping seizure foci and tumor genetic factors in glioma associated seizure patients. J Neurosurg Sci 2020;64:456-463.

9. Could the authors publish their data in Mendeley data?

https://data.mendeley.com/

Author Response

Dear reviewer:

    We would like to thank you for your careful reading, helpful comments, and constructive suggestions, which has significantly improved the presentation of our manuscript.We have carefully considered all comments from the reviewers and revised our manuscript accordingly.

Round 3

Reviewer 2 Report

The reviewer could not find the Institutional Review Board Number of the present manuscript. This is mandatory for clinical research. It should be added to the methodology and "Institutional Review Board Statement."